# Effect of Treatment in a Specialized Pediatric Hemato-Oncology Setting on 5-Year Survival in Acute Lymphoblastic Leukemia: A Quasi-Experimental Study

**DOI:** 10.3390/cancers14102451

**Published:** 2022-05-16

**Authors:** Margrietha van der Linde, Nikki van Leeuwen, Frank Eijkenaar, Anita W. Rijneveld, Rob Pieters, Henrike E. Karim-Kos

**Affiliations:** 1Department of Public Health, Center for Medical Decision Making, Erasmus University Medical Center, 3015 GD Rotterdam, The Netherlands; n.vanleeuwen.1@erasmusmc.nl; 2Erasmus School of Health Policy and Management, Erasmus University Rotterdam, 3062 PA Rotterdam, The Netherlands; eijkenaar@eshpm.eur.nl; 3Department of Hematology, Erasmus University Medical Center, 3015 GD Rotterdam, The Netherlands; a.rijneveld@erasmusmc.nl; 4Princess Máxima Center for Pediatric Oncology, 3584 CS Utrecht, The Netherlands; r.pieters@prinsesmaximacentrum.nl (R.P.); h.e.karim-kos@prinsesmaximacentrum.nl (H.E.K.-K.); 5University Medical Center Utrecht, 3584 CX Utrecht, The Netherlands; 6Department of Research and Development, Netherlands Comprehensive Cancer Organization (IKNL), 3511 DT Utrecht, The Netherlands

**Keywords:** ALL, specialized pediatric hemato-oncology care, site of treatment, AYAs, 5-year survival, regression discontinuity, The Netherlands, causal inference

## Abstract

**Simple Summary:**

Adolescents and young adults (AYAs) with acute lymphoblastic leukemia (ALL) have a worse prognosis than children. In addition to differences in biology—such as higher incidence of unfavorable genetic alterations in the AYA population—this might be related to the fact that ALL patients under a certain age (often 18 years) are generally treated in special pediatric hemato-oncology settings, which is associated with improved survival, while patients above that age are treated in adult hemato-oncology care settings. Based on previous research, adult treatment settings have increasingly adopted pediatric-inspired protocols, which appear to have led to increased survival of adolescent ALL patients. This study aims to assess whether there remains an effect of treatment of ALL patients in a specialized pediatric hemato-oncology setting on 5-year survival. This study provides insight into the effects of such treatment for ALL patients, and may stimulate further research into causal relationships in other oncological conditions.

**Abstract:**

Survival rates of adolescents and young adults (AYAs) with acute lymphoblastic leukemia (ALL) are inferior to those of pediatric ALL patients. In part, this may be caused by differences in treatment setting. Generally, children are treated in specialized pediatric hemato-oncology settings, whereas AYAs are treated in adult hemato-oncology settings. Since 2005, adult treatment protocols have included pediatric-inspired chemotherapy, which has been the standard of care for AYAs from 2008 onwards. This study aims to assess whether, despite protocols in both settings having become more similar, there remains an effect of treatment in specialized pediatric hemato-oncology settings on 5-year survival for ALL patients in the Netherlands. We used nationwide registry data (2004–2013) on 472 ALL patients aged between 10 and 30 years old. A fuzzy regression discontinuity design was applied to estimate the treatment effect using two-stage least squares regression with the treatment threshold at 17 years and 7 months of age, adjusting for sex, age at diagnosis, and immunophenotype. We found a risk difference of 0.419 (*p* = 0.092; 95% CI = −0.0686; 0.907), meaning a 41.9 percentage point greater probability of surviving five years after diagnosis for ALL patients treated in specialized pediatric hemato-oncology settings. Our results suggest that ALL patients around the threshold could benefit from increased collaboration between pediatric and adult hemato-oncology in terms of survival.

## 1. Introduction

The most common form of pediatric cancer is acute lymphoblastic leukemia (ALL), which is also the most common cause of cancer-related mortality under the age of 20 years [1,2]. Survival estimates for ALL decline rapidly with age, with higher survival rates for pediatric ALL patients compared to adolescent and young adult (AYA) ALL patients [3]. In addition to differences in biology—such as higher incidence of unfavorable genetic alterations in the AYA population [4]—this difference in survival may be caused by differences in treatment setting. Typically, children are treated in specialized pediatric settings, whereas AYAs are treated in specialized adult settings [5,6]. The age limit for being eligible for treatment in specialized pediatric oncology settings differs between countries. In the Netherlands, it has been the intention since 2002 that all patients under 18 are treated in specialized pediatric oncology settings, although in practice this age threshold is not always strictly applied [7].

Approximately 15 years ago, several observational studies found an association between treatment with pediatric protocols and beneficial outcomes in AYA ALL patients [8,9,10]. In the Netherlands, findings by de Bont et al. [9] led to an alteration in treatment protocols for AYAs with ALL up to the age of 40. Specifically, since 2005, adult treatment protocols have included pediatric-inspired chemotherapy protocols. From 2008 onwards, this approach has been the standard of care in this age group. Dinmohamed et al. [11] showed that this approach considerably increased survival rates among 18–39-year-old ALL patients.

A major challenge in studies using observational data to estimate treatment effects is dealing with confounding by indication [12]. This refers to bias due to unmeasured differences in patient characteristics between the treatment and control groups. Incomparability between patients with and without exposure to the intervention under study (in this study, being treated in a specialized pediatric hemato-oncology setting) hampers causal inference between treatment and outcome. Experts in this field have therefore argued for the application of innovative (statistical) methods to handle confounding by indication in observational data [13,14,15,16,17]. Using nationwide registry data covering a 10-year period, the aim of this study is to assess whether, despite the fact that the protocols in both settings have become more similar, there remains an effect of treatment in a specialized pediatric hemato-oncology setting on 5-year survival for ALL patients in the Netherlands. For this purpose, we adopt a quasi-experimental approach utilizing sophisticated statistical methods that allow for causal interpretation of treatment effects.

## 2. Materials and Methods

### 2.1. Data Sources

We used linked data from the population-based Netherlands Cancer Registry (NCR) and the Dutch Childhood Oncology Group (DCOG) for the period 2004–2013 [7]. The data from the DCOG were used to establish the treatment setting. All newly diagnosed malignancies with classification according to the ICCC-3 (International Classification of Childhood Cancer, 3rd edition) are reported directly to the NCR by the Automated Pathological Archives (PALGA), and the registry is enhanced with reports from the National Registry of Hospital Discharges (LBZ) and hematology laboratories [18]. The data include follow-up until 1 February 2020, obtained through annual linkage with the nationwide Personal Record Database (BRP) containing vital status on all Dutch residents. From this dataset, we selected patients diagnosed with ALL (ICD-O-3 M9811-9818 and M9835-9837) and aged between 10 and 30 years. In addition, we only selected patients with complete follow-up (≥5 years); 0.6% of the patients were lost to follow-up. This led to the selection of 472 patients.

### 2.2. Variables

The primary outcome in this study was 5-year observed overall survival. Age at diagnosis, sex, and immunophenotype were identified as potential confounding variables, as they have been found to be associated with both treatment in specialized pediatric hemato-oncology care settings and 5-year survival [2,7,8,9]. Data on genetic abnormalities, which are a known confounder in the relationship between site of treatment and survival, were not available in the NCR. This aspect is further debated in the Discussion.

### 2.3. Design

In studies estimating treatment effects using observational data, bias is likely, due to confounding by indication [12,19]. Specifically, in the presence of confounding by indication, the assignment for treatment is strongly related to the characteristics of the patient and/or the preferences of the physician, which are potentially also associated with the outcome of interest. These characteristics and preferences can be both measured and unmeasured confounders. With standard regression methods, it is impossible to sufficiently correct for unmeasured confounding. In this study, we address this issue by using a regression discontinuity (RD) design to determine the effect of being treated in a specialized pediatric hemato-oncology setting on 5-year survival, enabling causal inference.

The RD design is an increasingly used quasi-experimental design in healthcare and elsewhere to investigate the causal effects of interventions [13,14,15]. It has been described as the next-best evaluation method after a randomized controlled trial (RCT) [14], and its potential in healthcare to provide evidence on treatment effects is illustrated in several recent publications [20,21,22,23,24]. The RD design is attractive as it allows for causal interpretation of estimated effects of interventions or exposures using real-world observational data. Specifically, the RD design allows for causal inference by exploiting treatment assignment rules [25]. Unlike in RCTs, in an RD design the treatment is not randomly assigned, but instead determined by a value lying on either side of a baseline assignment variable. A key assumption is that patients close to the treatment threshold are similar in baseline characteristics and, thus, interchangeable, much like treatment and control patients in RCTs. This assumption implies that no unobserved confounders are present in patients around the threshold [25,26]. By comparing outcomes between patients at both sides of the threshold, the RD design can provide causal effect estimates from observational data that are largely free from both measured and unmeasured confounding.

In the Netherlands, it has been the intention since 2002 to treat patients below the age of 18 years diagnosed with cancer in specialized pediatric oncology settings. However, the age of 18 did not appear to be a likely treatment assignment threshold in our data, as a relatively large proportion of 17-year-olds are not treated in specialized pediatric oncology settings [7]. Based on an exploratory analysis, we set the treatment assignment threshold at 17 years and 7 months. This threshold rule was used to assign patients to the treatment or control groups. To deal with the imperfect compliance with the treatment threshold, a fuzzy RD design was used. In contrast to a sharp RD design, in which treatment is assigned deterministically by the baseline assignment variable being below or above the threshold (i.e., the probability of treatment being either 0 or 1), the fuzzy RD design assigns treatment probabilistically, meaning that treatment status does not change for 100% at the threshold [27]. In other words, fuzzy RD exploits discontinuity in the probability of treatment assignment.

### 2.4. Statistical Analysis

A common approach to estimate a treatment effect in a fuzzy RD design is to use instrumental variable (IV) analysis, where the threshold rule functions as the instrument [27]. We used this approach in a 2-stage least squares (2SLS) regression analysis. Specifically, we combined two linear probability models (LPMs). In the first stage, we predicted the probability of being treated in specialized pediatric hemato-oncology care from the treatment threshold (i.e., yes/no—age below 17 years and 7 months at diagnosis), adjusting for the covariates age at diagnosis, sex, and immunophenotype. The second stage regressed the predicted treatment probability from the first stage (i.e., the instrument) on the outcome (yes/no—5-year survival), adjusting for the same covariates. The reason for adjusting for the same covariates in both stages was that prior work shows that these factors could influence both treatment assignment in a specialized pediatric hemato-oncology setting and 5-year survival [2,7,8,9]. In other words, this approach allows for adjustment for possible confounding of the relationship between instrument and outcome [28].

The second stage of the 2SLS regression yields an estimate of the effect of interest, i.e., the effect of treatment in a specialized pediatric hemato-oncology setting on the probability of surviving for 5 years after diagnosis among adolescent ALL patients. This effect is commonly referred to as the risk difference [28]—in this case, the percentage point difference in the probability of being alive 5 years after diagnosis in the treatment group relative to the control group. The risk difference is the effect of the intervention among those whose treatment assignment is determined by the threshold (called ‘compliers’), which should be interpreted as a local effect around the treatment threshold. In the literature, this effect is also referred to as the complier average causal effect (CACE) [27]. We interpret this effect to apply for patients present around the treatment assignment threshold, meaning slightly below or slightly above the age of 17 years and 7 months.

We assessed the relevance and strength of our instrument using the F-statistic which, according to previous works, should exceed 10 for a sufficiently strong instrument [29,30]. All statistical analyses were performed using R software version 4.0.3 (R Foundation for Statistical Computation, Vienna, Austria) using the estimatr and rms packages.

## 3. Results

### 3.1. Descriptive Statistics of the Study Sample

The treatment group (aged from 10 years to 17 years and 7 months) and control group (aged from 17 years and 7 months to 30 years) consisted of 284 and 188 patients, respectively (Table 1). Almost two-thirds (63%) of the total study sample were male, and the median age at diagnosis was 16 years. Overall, 59% of the patients were treated in a specialized pediatric hemato-oncology care setting. The probability of being treated in a specialized pediatric hemato-oncology setting in the treatment group was 97%, while it was 1.1% in the control group. The overall unadjusted 5-year survival rate was 76%; this rate was considerably higher for the treatment group than for the control group: 82% versus 68%. Approximately 74% and 72% of the treatment and control groups had the BCP-ALL immunophenotype, respectively. 

Figure 1 shows the probability of being treated in a specialized pediatric hemato-oncology setting (Figure 1A) for the treatment and control groups, where the imperfect compliance with the treatment threshold (see Section 2.3) is clearly visible. The five-year survival rate (Figure 1B) shows more variation in the control group than in the treatment group, which appears to be related to the smaller number of patients (Figure 1C).

### 3.2. Estimation Results

The 2SLS regression analysis yielded an estimate of 0.419, which is statistically significant at a 10% level (*p* = 0.092; 95% CI = −0.069; 0.91). This effect can be interpreted as a 41.9 percentage point greater probability of being alive after five years for patients treated in specialized pediatric settings compared to specialized adult settings. The F-statistic of our instrument was 678 (*p* < 0.001), indicating a sufficiently strong instrument.

## 4. Discussion

### 4.1. Summary and Discussion of Main Findings

In this study, we assessed the causal effect of treatment in a specialized pediatric hemato-oncology setting on the probability of surviving for five years after diagnosis for ALL patients in the Netherlands. Applying a quasi-experimental fuzzy RD design on nationwide registry data, we found a 41.5 percentage point greater chance of 5-year survival for ALL patients treated in a specialized pediatric hemato-oncology setting. This effect was statistically significant at a 10% significance level. Our results suggest that it could be beneficial for the survival of AYA ALL patients around the cutoff if closer collaboration between pediatric and adult hemato-oncologists is encouraged, which is consistent with previous findings from observational studies [8,9,10]. Additionally, the therapeutic landscape for AYA ALL—and, thus, the collaboration between pediatric and adult hemato-oncology centers—may change significantly in the near future due to the development of immunotherapeutic strategies. Examples of these therapies are bispecific antibodies (such as blinatumomab) and antibody drug conjugates (such as inotuzumab) first studied in adult ALL, and CAR-T cells first studied in pediatric ALL.

As a consequence of using the RD design, we cannot distinctly define the age of the patients to whom the treatment effect applies. The risk difference must be interpreted as a local treatment effect estimate that applies to patients whose assignment variable is close to the threshold [27]. In our study, this means that the closer patients are to the treatment assignment threshold of age 17 years and 7 months, the more likely it is that the treatment effect applies. The estimated effect does not necessarily apply to patients with an assignment variable (age) farther away from the threshold, since these patients may not be as comparable to one another as treatment and control patients closer to the threshold.

Despite reduced differences between the treatment protocols of specialized pediatric and specialized adult hemato-oncology settings during our study period, we still found a substantial difference in survival. Below, we discuss three potential explanations for this: (1) differences between pediatric and adult treatment protocols in terms of risk stratification; (2) execution of the treatment protocols; and (3) lower participation rates in clinical trials among AYAs than among children, which may affect survival [31,32].

In both pediatric and adult hematology settings, patients are assigned to treatments of varying intensity based on patient and tumor characteristics [33]. However, the risk stratification of both settings seems to differ in important respects, which may result in patients with similar characteristics (other than age) receiving different treatments. First, the treatment protocols show differences in drug selection and dose intensity. Specifically, in specialized pediatric settings, patients without high-risk factors are eligible for chemotherapy with reduced intensity in order to achieve a favorable treatment outcome with as few adverse effects as possible. This element was absent or less present in adult protocols at that time. Patients with high-risk factors (for example, genetic abnormalities in leukemia cells) but good minimal residual disease (MRD) response—which is an important indicator of favorable therapy response—received stem cell transplantation (SCT) in specialized adult hematology settings, whereas they received maintenance chemotherapy in specialized pediatric hemato-oncology settings [34,35]. Today, patients without high-risk genetic abnormalities and MRD negativity are no longer offered allogeneic SCT. Importantly, SCT is associated with higher treatment-related mortality (15–22%) [36]. In the past, non-randomized studies in adult ALL showed that treatment according to adult chemotherapy protocols led to worse outcomes than SCT. However, two recent studies suggest that contemporary pediatric-inspired chemotherapy protocols lead to improved survival compared to SCT in adults with ALL up to the age of 50 years [37,38].

One of the differences related to execution of the protocols might be non-compliance—treatment protocols are more likely to be discontinued in adult hemato-oncology settings than in pediatric hemato-oncology settings, which may be related to treatment-related toxicity [39,40]. More intensive treatment increases the chance on treatment-related toxicity among AYAs. Older adolescents experience more hematological as well as acute toxicity compared to children. As suggested by Reedijk et al. [2], there are differences in how this treatment-related toxicity is managed between the adult and pediatric settings.

A known hypothesis for the difference in survival between children and AYAs with ALL is the lower enrollment of AYAs in clinical trials [31,32], recently acknowledged by the term *Accrual Cliff* [41]. Participation in clinical trials has been shown to improve outcomes in AYA ALL [42,43]. Although clinical trial participation among AYAs has increased over the past few decades [44], disparities remain, which may be part of the explanation for the survival difference that we found.

### 4.2. Strengths and Limitations

This is the first study to examine the potential causal relationship between treatment in specialized pediatric hemato-oncology care and 5-year survival among adolescent ALL patients. For this purpose, we were able to use nationwide registry data on all Dutch patients who were diagnosed with ALL during a 10-year period. A previous work showed that 96% of all patients diagnosed with cancer in the Netherlands are included in this registry [45]. The registration data we used for this study were of very high quality, as also indicated by the low percentage of incomplete data (0.6% were lost to follow-up). 

Another strength of this study lies in its use of the RD design. Unlike the frequently applied method of survival analysis, the RD design allows for causal inference, although Bor et al. [21] have proposed a method that combines fuzzy RD with survival analysis, which we consider to be an interesting methodology for follow-up research.

However, the validity of RD strongly relies on the assumption that patients close to the threshold have similar (un)observed characteristics [27], much like treatment and control patients in RCTs. If this assumption is true, the estimated effect can be interpreted as causal. It is a well-known fact that AYA ALL patients have more frequent unfavorable genetic abnormalities compared to young children, contributing to worse outcomes in the AYA population [4]. For example, there is a significantly higher percentage of AYA ALL patients with the BCR-ABL translocation and with BCR-ABL-like ALL. Simultaneously, there is a much lower incidence of AYA patients with the favorable ETV6-RUNX1 rearrangements and with high-hyperdiploid ALL [4,46]. The genetic abnormalities are of strong prognostic importance, and change with age, even within our AYA group of 10–30 years [4]. If patients near the threshold have different genetic abnormalities, this would be a violation of the RD assumptions. However, we do not expect this to explain the entire survival difference that we found, as the genetic abnormalities do not influence treatment assignment, as these are determined after referral. Additionally, the genetic abnormalities change gradually with age. In other words, it is reasonable to assume that patients slightly above and slightly below the treatment assignment threshold of 17 years and 7 months are similar in terms of underlying genetic abnormalities.

Furthermore, another potential violation of the RD design is related to the findings of Reedijk et al. [7], who found that boys are more often treated in specialized pediatric oncology care than girls, and suggested that in addition to sex itself—for which we could adjust—this might partially be explained by the often more mature appearance of girls. To the extent that appearance influences outcomes, this may be a limitation of our study.

Another limitation is the relative inefficiency of the RD design. To estimate reliable treatment effects, the RD design demands at least 2.75 times the number of patients compared to a conventional RCT [47]. Additionally, statistical power is often a problem with 2SLS methods; the variance of such estimates is much larger compared to OLS regression at a given sample size, as this method includes two regression stages [28]. It is likely that our study suffers from this power problem. For follow-up research, we therefore recommend a larger study population, which could be achieved by extending the study period rather than widening the age range, as the latter would reduce clinical relevance.

An important caveat is that our data were derived from 2004 to 2013; thus, our findings apply only to that time period. In the meantime, there have been changes in guidelines and treatment in both AYA and pediatric treatment—for example, the addition of PEGylated asparaginase (which has been shown to improve clinical outcomes and survival) instead of L-asparaginase (used before that time period) to the standard treatment regimen for AYA ALL patients, and the centralization of the pediatric oncology in the Princess Maxima Center for Pediatric Oncology in 2018. Therefore, we are unable to make any statements about the present situation.

A final limitation is that, given our data, we only analyzed 5-year overall survival. Quality of cancer care and quality of survival are much more extensive than simply studying survival, as late treatment effects—such as osteonecrosis due to glucocorticoids, and infertility due to conditioning regimens for SCT—are very important. In addition to using larger study populations, future research should therefore also assess the effects on other outcome measures.

## 5. Conclusions

For ALL patients around the treatment threshold of 17 years and 7 months, we found evidence of a positive causal relationship between treatment in a specialized pediatric oncology care setting and 5-year survival. Our findings suggest that ALL patients slightly below and slightly above the age of 17 years and 7 months could benefit from increased collaboration between pediatric and adult oncology in terms of 5-year survival. However, for older AYAs, we must stress that higher-risk disease and higher toxicity rates with increased doses might be confounding factors, in addition to the site of treatment. In the near future, the therapeutic landscape for AYA ALL, along with the collaboration between pediatric and adult hemato-oncology centers, should be further stimulated, and may change significantly due to the development of novel immunotherapeutic strategies.

## Figures and Tables

**Figure 1 cancers-14-02451-f001:**
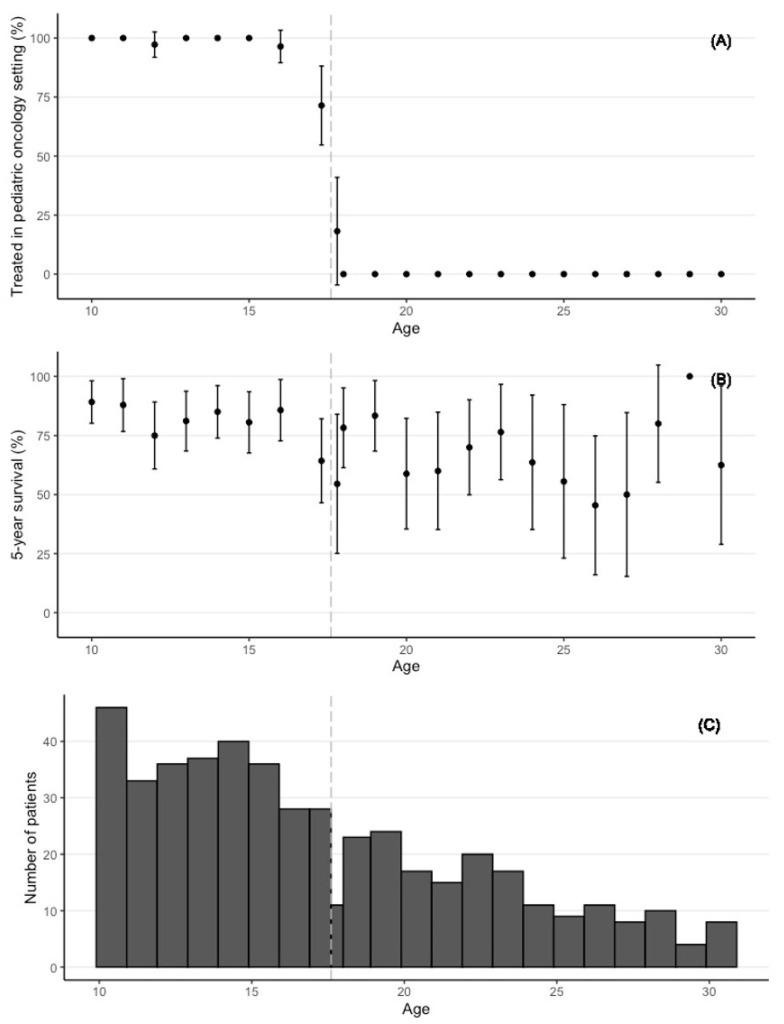
(**A**) The percentage of the patients by age that were treated in specialized pediatric hemato-oncology care. (**B**) Unadjusted 5-year survival as a percentage of patients by age. (**C**) Number of patients by age. In panels (**A**–**C**), the vertical dashed lines represent the treatment threshold at 17 years and 7 months.

**Table 1 cancers-14-02451-t001:** Descriptive statistics of patients aged 10–30 years with acute lymphoblastic leukemia, 2004–2013.

	Total	10 Years–17 Years and 7 Months (Treatment Group)	17 Years and 7 Months–30 Years (Control Group)
**N (%)**	**472 (100)**	**284 (100)**	**188 (100)**
Treatment setting			
Pediatric hemato-oncology [N (%)]	276 (58.5)	274 (96.5)	2 (1.1)
Adult hemato-oncology [N (%)]	196 (41.5)	10 (3.5)	186 (98.9)
Alive 5 years after diagnosis [N (%)]	360 (76.3)	232 (81.7)	128 (68.1)
Age at diagnosis (years)			
Median (interquartile range)	16 (13–20)	13 (11–15)	22 (19–25)
**Sex**			
Male [N (%)]	296 (62.7)	180 (63.4)	116 (61.7)
Female [N (%)]	176 (37.3)	104 (36.6)	72 (38.3)
Immunophenotype *			
BCP-ALL [N (%)]	346 (73.3)	211 (74.3)	135 (71.8)
T-cell ALL [N (%)]	126 (26.7)	73 (25.7)	53 (28.2)

* BCP-ALL: B-cell precursor acute lymphoblastic leukemia [2].

## Data Availability

The data that support the findings of this study are not publicly available, and restrictions apply to the availability of the data used in the present study. Upon reasonable request, and with permission of the Netherlands Comprehensive Cancer Organization and the Princess Máxima Center for Pediatric Oncology, data can be made available by the last author.

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
