# Peer review of "Effect of Treatment in a Specialized Pediatric Hemato-Oncology Setting on 5-Year Survival in Acute Lymphoblastic Leukemia: A Quasi-Experimental Study"

_cancers, 2022, doi:10.3390/cancers14102451_

Round 1
Reviewer 1 Report
This study was designed to evaluate, through a specific and innovative statistical method and by exploiting reliable data available from a national repository in The Netherlands (period 2004-2013), the possible impact, on the 5-year survival of ALL, of treatment delivered in dedicated settings, either to pediatric or adult, in 472 patients aged 10-30 years.
They found a higher probability of surviving 5 years after diagnosis
for ALL patients treated in specialized pediatric hemato-oncology centers.
Authors should be commended for the effort produced in describing once more, through a specific statistical analysis, the different results obtained in the ALL adult and pediatric setting. This report is certainly of great value for the readers since further pushes towards an increased collaboration between pediatric and adult hemato-oncology to improve the results in AYA with ALL.
The paper is very well written with the evaluation plan and the statistical methods clearly explained and fully understandable also for readers with limited statistical skills. Results are well described and the discussion is characterized by scientifically valid arguments and good sense The authors acknowledge in the discussion also the weaknesses of the study (especiailly the study period) but this reviewer believes that the paper has a high relevance in the field of ALL treatment in the settings studied. The paper does not need major modifications
Author Response
Response to Reviewer 1 Comments
This study was designed to evaluate, through a specific and innovative statistical method and by exploiting reliable data available from a national repository in The Netherlands (period 2004-2013), the possible impact, on the 5-year survival of ALL, of treatment delivered in dedicated settings, either to pediatric or adult, in 472 patients aged 10-30 years.
They found a higher probability of surviving 5 years after diagnosis for ALL patients treated in specialized pediatric hemato-oncology centers.
Authors should be commended for the effort produced in describing once more, through a specific statistical analysis, the different results obtained in the ALL adult and pediatric setting. This report is certainly of great value for the readers since further pushes towards an increased collaboration between pediatric and adult hemato-oncology to improve the results in AYA with ALL.
The paper is very well written with the evaluation plan and the statistical methods clearly explained and fully understandable also for readers with limited statistical skills. Results are well described and the discussion is characterized by scientifically valid arguments and good sense The authors acknowledge in the discussion also the weaknesses of the study (especially the study period) but this reviewer believes that the paper has a high relevance in the field of ALL treatment in the settings studied. The paper does not need major modifications.
Response: Thank you for this positive feedback and for your time and effort in reviewing our manuscript.

Reviewer 2 Report
The present article focuses on a quasi-experimental study through which a prevalent problem is objectively addressed, which is the outcome in the therapeutic treatment of patients with acute lymphoblastic leukemia whose scheme changes by age range between children and adolescents/young adults. Certainly there have been numerous publications on the subject, but in this occasion it has been demonstrated through a regression discontinuity design, the importance of carefully considering the variables that affect the therapeutic decision making for patients who are especially in the limits that demarcate the age ranges.
The results are not shocking, but they do show what has been pragmatically assumed. The authors give a very clear account of the limitations of the study and I consider that a complete and objective discussion has been made. I consider that the article can be published in its current version, only with a revision of the text to avoid errors such as the one in line 28, which reads "lymphatic" where it should read "lymphoblastic".
Author Response
The present article focuses on a quasi-experimental study through which a prevalent problem is objectively addressed, which is the outcome in the therapeutic treatment of patients with acute lymphoblastic leukemia whose scheme changes by age range between children and adolescents/young adults. Certainly there have been numerous publications on the subject, but in this occasion it has been demonstrated through a regression discontinuity design, the importance of carefully considering the variables that affect the therapeutic decision making for patients who are especially in the limits that demarcate the age ranges.
The results are not shocking, but they do show what has been pragmatically assumed. The authors give a very clear account of the limitations of the study and I consider that a complete and objective discussion has been made.
Point 1: I consider that the article can be published in its current version, only with a revision of the text to avoid errors such as the one in line 28, which reads "lymphatic" where it should read "lymphoblastic".
Response 1: This was a mistake, thank you for noticing. We changed "lymphatic" into "lymphoblastic" (line 28). We have critically reviewed the entire manuscript for these types of errors and, as requested, we used the English editing service of MDPI.

Reviewer 3 Report
Dear Authors,
Thank you for your contribution. I read your article about the causal effect of treatment in a specialized pediatric hemato-oncology setting on the probability of surviving 5 years after diagnosis for acute lymphoblastic leukemia patients in the Netherlands. I think it is a relevant topic since this relationship is still not exanimated. I found interesting your method to address this issue by using a regression discontinuity (RD) design, that I found described in the paper quite well. My only concern is that the limitations of the study you detailed in the specific session, especially the potential impact of the genetic abnormalities on the survival, may impact the consistence of your analysis.
My comments, mostly minor, after reviewing are the following:
- Materials and methods:
- Line 85. Please add the explanation of the abbreviation ICCC-3 (International Classification of Childhood Cancer, 3rd edition)
- Line 114. In the sentence ‘the next best thing’ I would change the word thing with a more specific one, such as evaluation method
- Line 154. ‘In other words:..’ I would put a comma instead of colon
- Results:
- Line 178. As far as I understand from the Table 1, the probability of being treated in a specialized pediatric hemato-oncology setting in the control group is 1.1% and not 2%
- Line 179. Is there a statiscal significance difference in 5-year survival rate beetween the two groups (82% versus 68%)?
Kind regards
Author Response
Response to Reviewer 3 Comments
Dear Authors,
Thank you for your contribution. I read your article about the causal effect of treatment in a specialized pediatric hemato-oncology setting on the probability of surviving 5 years after diagnosis for acute lymphoblastic leukemia patients in the Netherlands. I think it is a relevant topic since this relationship is still not exanimated. I found interesting your method to address this issue by using a regression discontinuity (RD) design, that I found described in the paper quite well.
Point 1: My only concern is that the limitations of the study you detailed in the specific session, especially the potential impact of the genetic abnormalities on the survival, may impact the consistence of your analysis.
Response 1: Thank you, we agree that this is a concern, but unfortunately we simply did not have the data to adequately address this. We therefore addressed this issue as a limitation in the discussion section (lines 317-330). We believe we have elaborated on this particular issue sufficiently in the article.
Point 2: Line 85. Please add the explanation of the abbreviation ICCC-3 (International Classification of Childhood Cancer, 3rd edition)
Response 2: Thank you, we agree and added the explanation of the abbreviation (line 85).
Point 3: Line 114. In the sentence ‘the next best thing’ I would change the word thing with a more specific one, such as evaluation method
Response 3: Thank you for this suggestion, we agree and changed “thing” into “evaluation method” (line 114).
Point 4: Line 154. ‘In other words:..’ I would put a comma instead of colon
Response 4: Thank you for this suggestion, we adjusted this in the manuscript (line 154).
Point 5: Line 178. As far as I understand from the Table 1, the probability of being treated in a specialized pediatric hemato-oncology setting in the control group is 1.1% and not 2%
Response 5: This was a mistake, thank you for noticing. We adjusted this probability in the text (line 178).
Point 6: Line 179. Is there a statistically significance difference in 5-year survival rate between the two groups (82% versus 68%)?
Response 6: Yes, based on a Chi-square test this difference in unadjusted 5-year survival is statistically significant between the two groups (p <.001). However, as Table 1 reports crude, unadjusted descriptive statistics, we believe adding this does not add additional value to our research.
